# Congruent Validity of Resting Energy Expenditure Predictive Equations in Young Adults

**DOI:** 10.3390/nu11020223

**Published:** 2019-01-22

**Authors:** Francisco J. Amaro-Gahete, Guillermo Sanchez-Delgado, Juan M.A. Alcantara, Borja Martinez-Tellez, Victoria Muñoz-Hernandez, Elisa Merchan-Ramirez, Marie Löf, Idoia Labayen, Jonatan R. Ruiz

**Affiliations:** 1Departament of Medical Physiology, School of Medicine, University of Granada, 18071 Granada, Spain; 2PROmoting FITness and Health through physical activity research group (PROFITH), Department of Physical Education and Sports, Faculty of Sport Sciences, University of Granada, 18071 Granada, Spain; gsanchezdelgado@ugr.es (G.S.-D.); juanma.alcantara@hotmail.com (J.M.A.A.); borjammt@gmail.com (B.M.-T.); mariavmuher@gmail.com (V.M.-H.); elymerchan@hotmail.com (E.M.-R.); 3Department of Medicine, Division of Endocrinology, and Einthoven Laboratory for Experimental Vascular Medicine, Leiden University Medical Center, Leiden, Post Zone C7Q, P.O. Box 9600, 2300 RC Leiden, The Netherlands; 4Department of Biosciences and Nutrition, Karolinska Institutet, SE-141 83 Huddinge, Sweden; marie.lof@ki.se; 5Institute for Innovation & Sustainable Development in Food Chain (IS-FOOD), Public University of Navarra, 31006 Pamplona, Spain; idoia.labayen@unavarra.es

**Keywords:** metabolic rate, basal metabolism, indirect calorimetry, energy balance, obesity

## Abstract

Having valid and reliable resting energy expenditure (REE) estimations is crucial to establish reachable goals for dietary and exercise interventions. However, most of the REE predictive equations were developed some time ago and, as the body composition of the current population has changed, it is highly relevant to assess the validity of REE predictive equations in contemporary young adults. In addition, little is known about the role of sex and weight status on the validity of these predictive equations. Therefore, this study aimed to investigate the role of sex and weight status in congruent validity of REE predictive equations in young adults. A total of 132 young healthy adults (67.4% women, 18–26 years old) participated in the study. We measured REE by indirect calorimetry strictly following the standard procedures, and we compared it to 45 predictive equations. The most accurate equations were the following: (i) the Schofield and the “Food and Agriculture Organization of the United Nations/World Health Organization/United Nations” (FAO/WHO/UNU) equations in normal weight men; (ii) the Mifflin and FAO/WHO/UNU equations in normal weight women; (iii) the Livingston and Korth equations in overweight men; (iv) the Johnstone and Frankenfield equations in overweight women; (v) the Owen and Bernstein equations in obese men; and (vi) the Owen equation in obese women. In conclusion, the results of this study show that the best equation to estimate REE depends on sex and weight status in young healthy adults.

## 1. Introduction

The main component of daily energy expenditure (60%–70%) is resting energy expenditure (REE) [1]. Having valid and reliable REE estimations is crucial to establish reachable goals for dietary and exercise interventions. For the majority of clinics and nutrition centers, it is difficult to get REE measures through indirect calorimetry, because of time constraints and the high cost of the devices. Thus, REE predictive equations are commonly used as an alternative method [2]. REE predictive equations have been used in healthy and unhealthy people, young and old adults of different ethnicities, and in people with a wide range of body weight and body composition characteristics [2,3,4,5,6,7,8,9,10,11,12,13,14,15,16,17,18,19,20,21].

Indirect calorimetry (IC) is considered the reference REE measurement technique [22]. In order to minimise possible estimation errors, the method requires strict measurement conditions and a correct procedure to analyse the IC data [23,24]. REE predictive equations are used in clinical practice, but some of them are specific for certain population groups, including different weight status or different ethnic groups [4,6,16,17,25]. Two predictive equations for young adults were recently validated [4,11], yet little is known about the role of sex and weight status on the validity of these predictive equations. Moreover, most of the REE predictive equations were developed some time ago and, as the body composition of the current population has changed [26], it is highly relevant to assess the validity of REE predictive equations in contemporary young adults.

In the present study, we systemically reviewed the available REE predictive equations including information on age, height, weight, sex, fat mass, and fat free mass. Then, we compared the measured versus estimated REE. Therefore, the purpose of this study was to investigate the role of sex and weight status in congruent validity of REE predictive equations in young adults. 

## 2. Materials and Methods 

### 2.1. Participants

A total of 132 participants (67.4% female) aged 18–26 years old participated in the study. The participants were enrolled in the ACTIBATE study (ClinicalTrials.gov ID:NCT02365129) [27]. The participants reported being engaged in less than 20 minutes on 3 days/week of physical activity, having had a stable weight over the last three months (body weight changes <3 kg), being free of medications or diseases that might interfere with REE measurement, and not being enrolled in a weight-loss program. Each participant provided both oral and written informed consent prior to the initiation of study procedures. The study was in accordance with the Helsinki Declaration and was approved by the Human Research Ethics Committee of both the University of Granada (n° 924) and Servicio Andaluz de Salud (Centro de Granada, CEI-Granada).

### 2.2. Body Composition

Body weight (±10 g) and height (±0.1 cm) were measured using a digital integrating scale (SECA 760, Hamburg, Germany) and a stadiometer (SECA 220, Hamburg, Germany). Body mass index (BMI) was calculated as weight (kg)/height (m^2^). The participants were categorized as normal weight (BMI = 18.5–24.9 kg/m^2^), overweight (BMI = 25–29.9 kg/m^2^), and obese (BMI > 30 kg/m^2^) [28]. We determined fat mass and lean mass by dual energy X-ray absorptiometry (Discovery Wi, Hologic, Inc., Bedford, MA, USA).

### 2.3. Assessment of REE through Indirect Calorimetry

REE was measured through IC strictly following current methodological recommendations [23,24]. The participants were instructed not to engage in any physical activity 48 h before the measurement and to arrive by car or bus at 08.15 in a fasting condition of at least 12 h.

The participants were evaluated in a peaceful and relaxing environment where temperature (22.8 ± 0.9 °C) and humidity (43.6 ± 6.6%) were controlled. The participants lay on a bed in a supine position and were covered by a sheet. Then, they were instructed to breathe normally, and not to fidget, talk, or sleep. After 30 min of rest, respiratory exchange measurements were determined using a CCM Express system (Med graphics Cardiorespiratory Diagnostic, Saint-Paul, MN, USA) in 59 participants (37 women) and using a CPX Ultima CardiO2 system (Medical Graphics Corp, St Paul, MN, USA) in 73 participants (52 women). Both measurements required the use of a neoprene facemask, equipped with a directconnect™ metabolic flow sensor (Medgraphics Corp, Saint Paul, MN, USA). The measurements took 30 min, and we selected the most stable five-consecutive-minute periods (after discarding the first five minutes) for analysis (Breeze Software, MGC Diagnostic^®^, Breeze Suite 8.1.0.54 SP7) [29,30]. REE was calculated by the Weir abbreviated equation (assuming negligible protein oxidation) and expressed as Kcal/day [31], as follows: REE = [3.9 (VO2) + 1.1 (VCO2)] * 1.44.

### 2.4. REE Predictive Equations

We conducted a systematic search for publications reporting REE predictive equations in PubMed and Web of Science. We combined the following keywords in every possible combination: ‘energy metabolism’, ‘basal metabolism’, and ‘indirect calorimetry’, and additional terms (‘predict*’, ‘estimat*’, ‘equation*’, and ‘formula*’).

We only retrieved equations based on the following criteria: (i) performed in adults; (ii) based on weight, height, age, sex, and/or fat free mass and fat mass. The exclusion criteria included (i) equation derived from patients’ or athletes’ data, and (ii) small sample size (*n* < 50). A total of 45 predictive equations (see Appendix A) were included.

### 2.5. Statistical Analysis

We conducted analysis of covariance (ANCOVA) to compare measured (by IC) and predicted (by predictive equations) REE, adjusting for metabolic cart (i.e., CCM and MGC). We analysed the BIAS (mean error between measured minus predicted REE), the absolute differences (measured minus predicted in absolute terms), and the 95% limits of agreement. In order to classify participants’ under- or overprediction with every REE equation, we considered an accurate estimation when the equation predicted between 90% and 110% of the measured REE [32,33], considering underprediction and overprediction when the estimation was below 90% and above 110% of the measured REE, respectively. We also calculated the percentage of accurate prediction between 95% and 105% of the measured REE. We classified the participants into underprediction or overprediction when the estimation was below 95% and above 105% of the measured REE, respectively.

We also used the repeated measures analysis of variance (ANOVA) to determine differences among the study predictive equation that presented minor absolute differences and measured REE. The heteroscedasticity was examined using the Bland–Altman method [34], which plots the difference between predicted and measured REE versus the mean of predicted and measured REE. The analyses were conducted using the Statistical Package for Social Sciences (SPSS, v. 21.0, IBM SPSS Statistics, IBM Corporation, Armonk, NY, USA), and the level of significance was set at *p* < 0.05.

## 3. Results

The characteristics of the study sample are shown in Table 1. As expected, men had higher levels of lean mass and REE than women, whereas women had higher levels of fat mass (all *p* < 0.001).

Figure 1 shows the percentage of accurate prediction of REE predictive equations and the differences of the mean absolute values between predicted and measured REE in men by BMI categories. In normal weight men, the equations of Schofield [9] and the “Food and Agriculture Organization of the United Nations/World Health Organization/United Nations” (FAO/WHO/UNU) [5] provided 45% prediction accuracy (Figure 1A and Appendix A), 10% underpredictions, and 45% overpredictions (mean BIAS: −142 and −147 Kcal/day, respectively). There were no differences between the most accurate predictive equation (i.e., Schofield [9]) and the rest (Figure 1B, *p* = 0.303).

In overweight men, five equations [4,10,15,20,35] presented 41.7% of accurate predictions (Figure 1C and Appendix A). All of these equations overestimated REE (33%–50% of all participants). However, when we applied a severe accurate estimation (±5% of measured REE), the Livingston and Korth equations [13,20] provided the highest (33% and 25%, respectively) accurate predictions (mean BIAS was −183 and −69 Kcal/day, respectively). There were no differences across predictive equations (Figure 1D, *p* = 0.366). Also, there were no differences between the most accurate predictive equations (i.e., Livingston [13] and Korth [20]) and the rest (Figure 1D, *p* = 0.366).

In obese men, the Bernstein and Owen equations [14,16] provided 72.7% and 54.5% prediction accuracy (±10%), respectively. When applying a severe accurate estimation (±5% of measured REE), these equations presented 36.4% and 45.5% prediction accuracy, respectively (Figure 1E and Appendix A). The mean BIAS was 18 and 133 Kcal/day for the Bernstein and Owen predictive equations, respectively. We also observed significant differences (*p* = 0.007) when comparing the estimation of REE by the Bernstein and Owen equations [14,16] with the Muller predictive equation (which is the one that presented less accuracy) [8] (Figure 1F).

Figure 2 shows the percentage of accurate prediction of REE predictive equations, and the differences of the mean absolute values between predicted and measured REE in women by weight status categories. In normal weight women, several REE equations provided >60% prediction accuracy [5,9,13,18,19] (Figure 2A and Appendix A), yet when we applied a severe accurate estimation (±5% of measured REE), the Mifflin equation [19] showed the highest accuracy (40.7% prediction accuracy, mean BIAS: −50 Kcal/day). There were significant differences in the estimation of REE by the Mifflin equation [19] compared with the Korth [20], Weijs & Vansant [25], de la Cruz [11], Bernstein [16], and Owen [21] predictive equations (Figure 2B).

In overweight women, the Korth [20] and Johnstone [15] equations provided 61.9% and 57.1% prediction accuracy, respectively (Figure 2C and Appendix A). However, when taking into consideration a severe accurate estimation (±5%), the Korth equation presented 23.8% and the Johnstone equation 42.9% prediction accuracy (mean BIAS: −162 and −281 Kcal/day, respectively). There were significant differences between the Johnstone [15] and Frankenfield equations [36], and the Bernstein [16] and Owen equations [21] (both *p* = 0.002) (Figure 2D).

In obese women, the Owen equation [21] provided 66.7% prediction accuracy (mean BIAS: 68 Kcal/day) (Figure 2E and Appendix A). There were no differences between all REE predictive equations (*p* = 0.277).

Figure 3 shows Bland–Altman plots for the six selected equations in men and women and by weight status. The limits of agreement were large in all six cases: (i) −836 to 551 Kcal/day in normal weight men (using the Schofield equation [9], see Figure 3A and Appendix A), (ii) −465 to 364 Kcal/day in normal weight women (using the Mifflin equation [19], see Figure 3B and Appendix A), (iii) −998 to 631 Kcal/day in overweight men (using Livingston equation [13], see Figure 3C and Appendix A), (iv) −281 to 361 Kcal/day in overweight women (using the Johnstone equation [15], see Figure 3D and Appendix A), (v) −514 to 273 Kcal/day in obese men (using the Owen equation [14], see Figure 3E and Appendix A), and (vi) −333 to 470 Kcal/day in obese women (using the Owen equation [21], see Figure 3F and Appendix A). There was no interaction effect between any of the REE predicted equations, metabolic cart, and measured REE (all *p* > 0.2).

## 4. Discussion

The results of this study indicate that the most accurate equation to estimate REE differs by sex and weight status in young adults. The most accurate equations are (i) the Schofield [9] and FAO/WHO/UNU [5] equations in normal weight men; (ii) the Livingston [13] and Korth equations [20] in overweight men; (iii) the Owen [21] and Bernstein equations [16] in obese men; (iv) the Mifflin [19] and FAO/WHO/UNU [5] equations in normal weight women; (v) the Johnstone [15] and Frankenfield equations [32] in overweight women; and (vi) the Owen equation [21] in obese women. For practical purposes, we provide a flowchart decision tree to select an energy predictive equation by sex and weight status (see Figure 4).

### 4.1. Normal Weight Men

Our results show that the Schofield [9] and FAO/WHO/UNU equations [5] were the most accurate REE equations in young healthy adults, which is in line with previous studies [37,38,39]. The Schofield equation was derived from a sample of 7173 men and women, which included 4814 participants above the age of 18 and with a BMI between 21 and 24 kg/m^2^ (47% Italians). The FAO/WHO/UNU [5] equation was based on the Schofield equation [9] database and extended to 11,000 participants.

Willis et al. [4] recently proposed a predictive equation for young adults, and, although no differences were found between the Schofield and Willis equations, the first showed higher accuracy (45% vs. 40%, mean absolute differences: 280 ± 239 vs. 280 ± 222 Kcal/day, respectively). Of note is that participants’ characteristics in the study of Willis et al. [4] were substantially different compared with the participants enrolled in our study: (i) American versus Spanish population; (ii) BMI 28.7 ± 4.7 versus 22.4 ± 1.8 kg/m^2^, respectively; and (iii) REE 1866 ± 251 versus 1587 ± 390 Kcal/day, respectively.

### 4.2. Overweight Men

The results of our study showed that several equations (the Frankenfield, Johnstone, Korth, Livingston, Roza, and Willis equations) overpredicted REE (>50% of the study participants in all cases, mean absolute differences: 317 ± 256, 298 ± 226, 271 ± 205, 342 ± 273, 382 ± 330, and 325 ± 265 Kcal/day, respectively), and previous studies indicated similar trends when these equations were applied in other cohorts with similar characteristics [4,40]. In good agreement with other studies [2,25,32,33], we observed that the inclusion of body composition (fat free mass or fat mass) did not improve the accuracy of REE prediction in these participants, because Frankenfield, Johnstone, Livingston, Roza, and Willis did not include body composition variables. This is especially relevant because age-, weight-, and height-derived equations are more feasible in clinical practice.

### 4.3. Obese Men

A recent review conducted by an expert panel [32] concluded that the Mifflin equation [19] should be used for overweight and obese participants. Although this conclusion was based on limited data, another review confirmed that both the Mifflin and Owen equations [14,19] were accurate in overweight and obese participants [2]. We observed 45.5% prediction accuracy (mean absolute differences: 210 ± 174 Kcal/day) when we applied the Mifflin equation in our sample of obese men. Moreover, a study conducted in overweight and obese Dutch adults (BMI: 25–40 kg/m^2^) obtained higher accuracy when the Lazzer equation [7] was applied (almost 80% prediction accuracy), whereas we had 27.3% prediction accuracy when using this equation. This fact could be explained by the gases collection system and the gas analyser device used to determine REE [41]. In addition, we found 72.7% and 54.5% prediction accuracy (±10% and ±5% of measured REE, respectively) with the Bernstein equation [16]. Of note is that the participants’ characteristics in Bernstein’s study had a similar BMI compared with our study participants. Previous studies have shown that the estimation of REE is less accurate in obese than in non-obese subjects [25,32,33].

### 4.4. Normal Weight Women

Our results provide more evidence in the use of the Mifflin equation in normal weight women (61% prediction accuracy, mean absolute differences: 142 ± 157 Kcal/day). Our findings concur with other studies in non-obese individuals (aged 18–78, 82% prediction accuracy) [31], in normal weight European American women (63.7% prediction accuracy) [42], in overweight and obese individuals (aged 19–69 years old, 78% accurate prediction) [32], in extremely obese women (84% prediction accuracy) [43], and in overweight U.S. adults (almost 80% prediction accuracy) [2].

### 4.5. Overweight Women

In overweight women, the Korth equation (which includes fat free mass) showed the highest accuracy (61.9% prediction accuracy, mean absolute differences: 158 ± 119 Kcal/day). However, others [2,25,32,33] showed that the inclusion of fat free does not improve the accuracy of REE prediction. In our study, the Willis equation [4], which includes weight, age, and sex data of young adults (18–30 years old), showed 52% prediction accuracy (mean absolute differences: 137 ± 103 Kcal/day), which is considered acceptable [32]. However, the prediction accuracy observed in the study by Willis et al. [4] was 70%, which might be partially explained by the inclusion of normal weight and obese individuals as a part of the total sample included in their study.

### 4.6. Obese Women

Mifflin [19] is the recommended equation in obese women [42], as well as in individuals with different ages, BMI, and ethnicities [2,6,8,32,42]. Our results also revealed an acceptable prediction accuracy (44.4%, mean absolute differences: 176 ± 132 Kcal/day) of the Mifflin equation, but far from being the best predictive equation in this group. As observed in obese men, the highest accuracy equations in obese women were the Owen [14] and Bernstein equations [16] (66.7% and 55.6% prediction accuracy, respectively, and mean absolute differences: 136 ± 156 and 162 ± 166 Kcal/day, respectively). These results could be expected because the Owen equation usually underestimated the REE measure in young adults without considering BMI [4], and the Bernstein equation was proposed based on participants with the same characteristics as those in our study [16,21].

### 4.7. Limitations

Our study has some limitations. (i) The participants were young healthy adults, and we do not know if these results can be extended to older or unhealthy people. (ii) Our results for overweight men, obese men, and obese women need confirmation because of the small number of women and men in these groups; this fact could have influenced the results obtained, yet it is important to consider that our sample was more homogeneous than other studies conducted in young adults as a result of the strict inclusion criteria and the narrow age range [4]. (iii) The metabolic carts could overestimate or underestimate the REE measure, yet the data collection process and analysis were strictly controlled and standardised, which is certainly a strength. 

## 5. Conclusions

In conclusion, this study shows that there is a wide variation in the accuracy of REE predictive equations depending on sex and BMI index in young adults. Future studies are, however, needed to confirm the results obtained for overweight men, obese men, and obese women because of the relatively small sample size in our study. We provide a decision tree (Figure 4) to select an REE equation depending on sex and BMI in the individuals, taking into account the percentage of accurate prediction applying an accuracy of ±5% and ±5% of measured REE. We also provide an open access Excel sheet that automatically estimates REE using 47 equations considering sex, age, weight, and height, as well as individuals’ fat mass and/or fat free mass (if available) (see Appendix A) [44].

## Figures and Tables

**Figure 1 nutrients-11-00223-f001:**
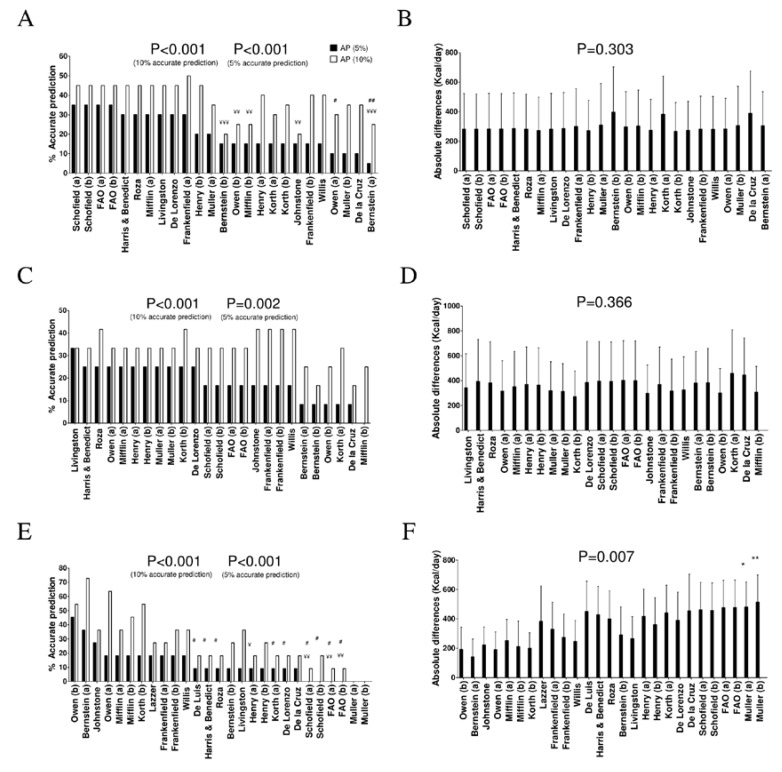
Percentage of accurate prediction of resting energy predictive equations and differences of mean absolute values between predicted and measured resting energy expenditure in men by weight status categories. (**A**) Percentage of accurate prediction at 5% and 10% of measured resting energy expenditure in normal weight men. (**B**) Mean (SD) absolute differences between predicted and measured resting energy expenditure in normal weight men. (**C**) Percentage of accurate prediction at 5% and 10% of resting energy expenditure measured in overweight men. (**D**) Mean (SD) differences between predicted and measured resting energy expenditure in absolute values in overweight young men. (**E**) Percentage of accurate prediction at 5% and 10% of resting energy expenditure measured in obese men. (**F**) Mean (SD) differences between predicted and measured resting energy expenditure in absolute values in obese men. (a) and (b) refer to predictive equations that are proposed by the same author, but require different anthropometry or body composition parameters. *p*-value of repeated measures analysis of variance (with Bonferroni post-hoc analysis) among the predictive equations. * *p* < 0.05; ** *p* < 0.01 when compared with the predictive equation that presented minor absolute differences with measured resting energy expenditure. ^¥^
*p* < 0.05; ^¥¥^
*p* < 0.01; ^¥¥¥^
*p* < 0.001 when compared with the predictive equation that presented the best resting energy expenditure accurate prediction (10%) with measured resting energy expenditure. ^#^
*p* < 0.05; ^##^
*p* < 0.01; ^###^
*p* < 0.001 when compared with the predictive equation that presented the best resting energy expenditure accurate prediction (10%) with measured resting energy expenditure. AP: accurate predictions. Abbreviations: FAO, “Food and Agriculture Organization of the United Nations/World Health Organization/United Nations” equation.

**Figure 2 nutrients-11-00223-f002:**
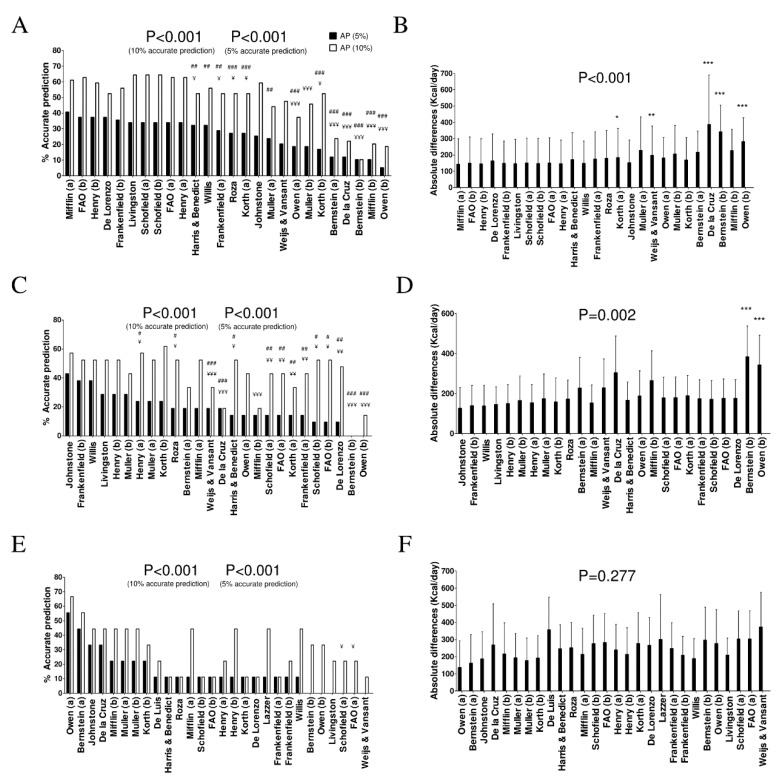
Percentage of accurate prediction of resting energy predictive equations and mean differences between predicted and measured resting energy expenditure in absolute values in women by weight status categories. (**A**) Percentage of accurate prediction at 5% and 10% of resting energy expenditure measured in normal weight women. (**B**) Mean (SD) differences between predicted and measured resting energy expenditure in absolute values in normal weight women. (**C**) Percentage of accurate prediction of several resting energy predictive equations at 5% and 10% of resting energy expenditure measured in overweight women. (**D**) Mean (SD) differences between predicted and measured resting energy expenditure in absolute values in overweight women. (**E**) Percentage of accurate prediction at 5% and 10% of resting energy expenditure measured in obese young women. (**F**) Mean (SD) differences between predicted and measured resting energy expenditure in absolute values in obese women. (a) and (b) refer to predictive equations that are proposed by the same author, but require different anthropometry or body composition parameters. *p*-value of repeated measures analysis of variance (with Bonferroni post-hoc analysis) among the predictive equations. * *p* < 0.05; ** *p* < 0.01; *** *p* < 0.001 when compared with the predictive equation that presented minor absolute differences with measured resting energy expenditure. ^¥^
*p* < 0.05; ^¥¥^
*p* < 0.01; ^¥¥¥^
*p* < 0.001 when compared with the predictive equation that presented the best resting energy expenditure accurate prediction (10%) with measured resting energy expenditure. ^#^
*p* < 0.05; ^##^
*p* < 0.01; ^###^
*p* < 0.001 when compared with the predictive equation that presented the best resting energy expenditure accurate prediction (10%) with measured resting energy expenditure. AP: accurate predictions.

**Figure 3 nutrients-11-00223-f003:**
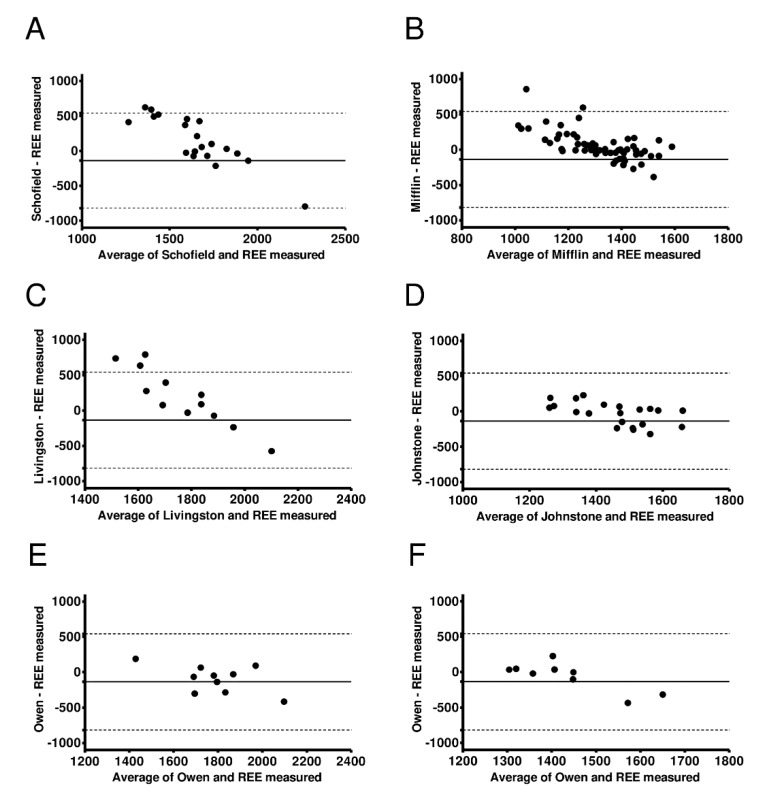
Bland–Altman plots for selected resting energy expenditure (REE) predictive equations. The solid lines represent the mean difference (BIAS) between predicted and measured REE. The upper and lower dashed lines represent the 95% limits of agreement. (**A**) normal weight young men; (**B**) normal weight young women; (**C**) overweight young men; (**D**) overweight young women; (**E**) obese young men; and (**F**) obese young women.

**Figure 4 nutrients-11-00223-f004:**
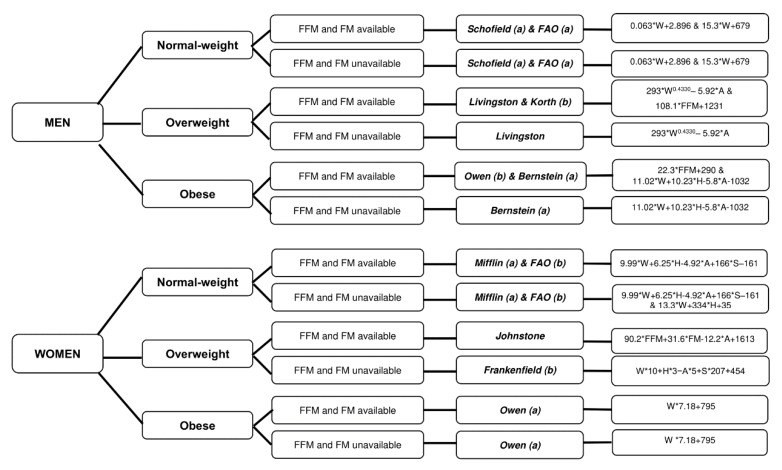
Decision tree to select a resting energy expenditure predictive equation by sex and weight status. (a) and (b) refer to predictive equations that are proposed by the same author but require different anthropometry or body composition parameters. Abbreviations: M: men; F: women; W: weight; H: height; A: age; S (men = 0; women = 1); FFM: fat free mass; FM: fat mass.

**Table 1 nutrients-11-00223-t001:** Descriptive parameters for the participants in the study.

	Men (*n* = 43)			Women (*n* = 79)		
	Normal weight (*n* = 20)	Overweight (*n* = 12)	Obese (*n* = 11)	Normal weight (*n* = 59)	Overweight (*n* = 21)	Obese (*n* = 9)
Age (years)	21.5 (2.0)	23.5 (2.1)	23.0 (2.5)	22.1 (2.1)	22.6 (2.4)	21.6 (2.0)
Weight (kg)	69.0 (7.6)	84.4 (7.6)	109.0 (10.5)	58.9 (7.1)	74.6 (6.5)	84.5 (9.7)
Height (m)	1.75 (0.06)	1.76 (0.06)	1.78 (0.06)	1.64 (0.07)	1.64 (0.06)	1.64 (0.09)
BMI (kg/m^2^)	22.4 (1.8)	27.1 (1.4)	34.5 (2.2)	21.8 (1.8)	27.6 (1.2)	31.3 (1.2)
Fat free mass (kg)	50.9 (5.4)	57.2 (4.1)	66.3 (6.5)	37.8 (4.1)	41.7 (4.0)	41.5 (6.0)
Fat mass (kg)	18.1 (4.8)	27.2 (6.8)	42.7 (6.4)	22.1 (4.5)	32.9 (3.6)	39.0 (5.0)
Fat mass (%)	24.9 (5.3)	30.6 (5.9)	37.8 (3.4)	35.2 (4.7)	43.0 (3.0)	45.1 (2.8)
REE (Kcal/day)	1587 (390)	1675 (363)	1870 (251)	1295 (222)	1481 (179)	1470 (203)

Data are expressed as mean (standard deviation). Abbreviations: BMI, body mass index; REE, resting energy expenditure.

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
