# Peer review of "Congruent Validity of Resting Energy Expenditure Predictive Equations in Young Adults"

_nutrients, 2019, doi:10.3390/nu11020223_

Reviewer 1 Report

 The study aims to assess the accuracy of different equations to predicting the REE in a population of young adults (normal weight, overweight, obese).

Several studies have evaluated the accuracy of these predictive equations in different population and currently there is poor clarity.

The following study even though it is well structured, unfortunately,  I think that the sample hasn't the number sufficiently adequate to make valid the aim of the study.

Please produce rationale for sample sizes, including clear power calculation for three goups.                                                                                                                       

Author Response

#Reviewer 1:

Comment 1: The study aims to assess the accuracy of different equations to predicting the REE in a population of young adults (normal weight, overweight, obese). Several studies have evaluated the accuracy of these predictive equations in different population and currently there is poor clarity. The following study even though it is well structured, unfortunately, I think that the sample hasn't the number sufficiently adequate to make valid the aim of the study. Please produce rationale for sample sizes, including clear power calculation for three groups. 

Authors’ reply 1: We appreciate the Reviewer’s summary. The determination of the sample size and power of the study were made based on the data of previous studies (Owen et al. 1986, Korth et al. 2007, Johnstone et al. 2006, and De la Cruz et al. 2014 among others). We considered resting energy expenditure differences between predictive equations in order to assess the sample size requirements for the one-way ANOVA. As a result, we expect to detect an effect size of 100 kcal considering a type I error of 0.05 with a statistical power of 0.80 if we consider a minimum of 10 participants per each sex and BMI category. Given the relatively small sample size in some cases, we have added as a limitation of our study that our results for overweight men, obese men, and obese women need future confirmation. However, it is important to consider that our sample was more homogeneous than other studies conducted in young adults as a result of the strict inclusion criteria and the narrow age range.

Reviewer 2 Report

I appreciate this invitation to review a manuscript by Francisco J. Amaro-Gahete et al. entitled: “Congruent validity of resting energy expenditure 2 predictive equations in young adults" submitted to the Nutrients.

Francisco J. Amaro-Gahete et al. conducted a small study to evaluate the role of sex and weight status in congruent validity of 45 REE predictive equations in 132 young adults. In spite of some limitations that have been listed in manuscript, and the necessity of further investigations, before the implication to the clinical practice, the conducted study shows that there is a wide variation in the accuracy of REE predictive equations, which depend on sex and BMI index.

The manuscript is well written. The study methods and procedures are well planned and performed, and show new interesting findings, especially provided the “decision tree” and Excel sheet. By the way, in manuscript it is written that this file is an open access, however I could not make any changes without a password, which was not included?

My only one question and doubt is about the inclusion criteria, especially if and how the potential hormonal/thyroid diseases were excluded, since the thyroid hormones may significantly influence a REE. It came to my mind while reading the results presented in Table 1. Please pay attention that the Obese Women presented slightly lower REE than Overweight Women, when actually with increasing body weight we should expect a higher REE levels? How it could be explained? Therefore I have a question if the possible impairment of REE due to a reduced peripheral effect of thyroid hormones was excluded? In the manuscript is written: “The participants reported (…) being free of medications or diseases that might interfere with REE measurement“, and the exclusion criteria that I found on the www.clinicaltrials.gov for the presented study mention in this regard only “Taking medication for thyroid”. Was it the only way to exclude any possible impact of thyroid hormones on the REE measurements? Because it could affect the results significantly, it should be well clarified and explained in manuscript.

Author Response

#Reviewer 2:

Comment 1: I appreciate this invitation to review a manuscript by Francisco J. Amaro-Gahete et al. entitled: “Congruent validity of resting energy expenditure predictive equations in young adults" submitted to the Nutrients. Francisco J. Amaro-Gahete et al. conducted a small study to evaluate the role of sex and weight status in congruent validity of 45 REE predictive equations in 132 young adults. In spite of some limitations that have been listed in manuscript, and the necessity of further investigations, before the implication to the clinical practice, the conducted study shows that there is a wide variation in the accuracy of REE predictive equations, which depend on sex and BMI index.

Authors’ reply 1: We appreciate the Reviewer’s summary.

Comment 2: The manuscript is well written. The study methods and procedures are well planned and performed, and show new interesting findings, especially provided the “decision tree” and Excel sheet. By the way, in manuscript it is written that this file is an open access, however I could not make any changes without a password, which was not included?

Authors’ reply 2: We appreciate Reviewer´s comment. We have blocked a number of cells in the Excel sheet because these cells have some excel commands that allow the automatically estimation of resting energy expenditure using 47 equations considering sex, age, weight, and height as well as individuals’ fat mass and/or fat free mass (if available). In any case, if the Reviewer and/or the Editor consider that the Excel sheet should be unlocked, we can modify it.

Comment 3: My only one question and doubt is about the inclusion criteria, especially if and how the potential hormonal/thyroid diseases were excluded, since the thyroid hormones may significantly influence a REE. It came to my mind while reading the results presented in Table 1. Please pay attention that the Obese Women presented slightly lower REE than Overweight Women, when actually with increasing body weight we should expect a higher REE levels? How it could be explained? Therefore, I have a question if the possible impairment of REE due to a reduced peripheral effect of thyroid hormones was excluded? In the manuscript is written: “The participants reported (…) being free of medications or diseases that might interfere with REE measurement “, and the exclusion criteria that I found on the www.clinicaltrials.gov for the presented study mention in this regard only “Taking medication for thyroid”. Was it the only way to exclude any possible impact of thyroid hormones on the REE measurements? Because it could affect the results significantly, it should be well clarified and explained in manuscript.

Authors’ reply 3: We appreciate the Reviewer´s comment. One of the exclusion criteria was to be free of medication for thyroid as the Reviewer mentioned, thus a possible impact of thyroid hormones on the resting metabolic rate measurements should be excluded. However, to be sure of this fact, we have conducted a one-way ANOVA comparing T3 and T4 levels by sex and BMI categories in our study sample, and found no significant differences in any case.

                              Men

                             (n=43)

                            Women

                             (n=79)

P value

   Normal         weight         (n=20)

Overweight      (n=12)

     Obese

     (n=11)

  Normal          weight          (n=59)

Overweight      (n=21)

   Obese

    (n=9)

T3   levels (pg/ml)

3.431±0.361

3.589±0.312

3.562±0.349

3.217±0.374

3.420±0.377

3.537±0.540

0.134

T4   levels (pg/ml)

0.984±0.122

0.965±0.135

0.970±0.118

0.921±0.117

0.975±0.144

0.887±0.114

0.285

We did not find significant differences in resting metabolic rate between overweight and obese women (1481 [179] vs 1470 [203] kcal). The Reviewer is right about the fact that increasing body weight it should expect a higher resting metabolic rate level. However, as we can see in Table 1 the differences in body weight between overweight and obese women are explained by fat mass. Considering that fat mass is a non-metabolically active tissue, and that the fat free mass (which is a metabolically active tissue) was similar in overweight and obese women, we can conclude that the resting metabolic rate levels are consistent.

Round  2

Reviewer 1 Report

I accept the response of the authors.